# Sequence-Based Prediction of Protein Phase Separation: The Role of Beta-Pairing Propensity

**DOI:** 10.3390/biom12121771

**Published:** 2022-11-28

**Authors:** Pratik Mullick, Antonio Trovato

**Affiliations:** 1Department of Physics and Astronomy ‘Galileo Galilei’, University of Padova, 35121 Padova, Italy; 2Institut National de Recherche en Informatique et Automatique (INRIA), Centre National de la Recherche Scientifique (CNRS), Institut de Recherche en Informatique et Systèmes Aléatoires (IRISA), University of Rennes, 35042 Rennes, France; 3Department of Operations Research and Business Intelligence, Wrocław University of Science and Technology, 50-370 Wrocław, Poland; 4National Institute of Nuclear Physics (INFN), Padova Section, 35121 Padova, Italy

**Keywords:** protein droplets, liquid-liquid phase separation, amyloid aggregates

## Abstract

The formation of droplets of bio-molecular condensates through liquid-liquid phase separation (LLPS) of their component proteins is a key factor in the maintenance of cellular homeostasis. Different protein properties were shown to be important in LLPS onset, making it possible to develop predictors, which try to discriminate a positive set of proteins involved in LLPS against a negative set of proteins not involved in LLPS. On the other hand, the redundancy and multivalency of the interactions driving LLPS led to the suggestion that the large conformational entropy associated with non specific side-chain interactions is also a key factor in LLPS. In this work we build a LLPS predictor which combines the ability to form pi-pi interactions, with an unrelated feature, the propensity to stabilize the β-pairing interaction mode. The cross-β structure is formed in the amyloid aggregates, which are involved in degenerative diseases and may be the final thermodynamically stable state of protein condensates. Our results show that the combination of pi-pi and β-pairing propensity yields an improved performance. They also suggest that protein sequences are more likely to be involved in phase separation if the main chain conformational entropy of the β-pairing maintained droplet state is increased. This would stabilize the droplet state against the more ordered amyloid state. Interestingly, the entropic stabilization of the droplet state appears to proceed according to different mechanisms, depending on the fraction of “droplet-driving“ proteins present in the positive set.

## 1. Introduction

Liquid-liquid phase separation (LLPS), or demixing, with the coexistence of a diluted state and a dense condensed one, is a classic subject in polymer physics [1]. The essential physics is aptly captured within the Flory-Huggins (FH) approach, a simple lattice theory, where the free energy of mixing per lattice site can be derived using a mean-field assumption [2,3]. The driving force underlying LLPS is the exchange of chain/solvent interactions for chain/chain and solvent/solvent interactions under conditions for which this process is energetically favorable, a balance quantified by the Flory parameter.

Evidence has been mounting in the last years that protein LLPS underlies the formation of membrane-less organelles (MLOs) in living cells [4]. In fact, eukaryotic cells are composed of numerous compartments or organelles that carry out specific functions and provide spatio-temporal control over cellular materials, metabolic processes, and signaling pathways. However, cells also harbor several MLOs that lack a delimiting membrane. These are supra-molecular assemblies composed of proteins, nucleic acids, and other molecular components, that are present in the nucleus as well as in the cytoplasm. On the other hand, bacterial biomolecules were also shown to undergo LLPS [5], as well as viral ones [6].

One feature that has attracted considerable attention is the presence of intrinsically disordered regions (IDRs) in proteins that have the ability of driving LLPS [7]. These regions display a sequence-intrinsic preference for conformational heterogeneity or disorder under native conditions [8]. The detailed understanding of the biological function of disordered bio-molecular condensates, whose formation is driven by LLPS, is currently the focus of a major effort undertaken by a large community in cell biology [9].

In particular, several key proteins in neuro-degenerative disorders are components of MLOs [10]. The observed conversion of dynamic protein droplets to solid aggregates [11] shows them to be meta-stable or inherently unstable, and shows that specific cellular processes keep them from solidifying. These liquid-to-solid transitions are accelerated by disease mutations [12] that seem to target β-zippers in IDRs [13], which makes them more prone to fold into stable amyloid structures [14].

In fact, it was recently suggested that in the cell crowded environment the liquid condensed state should be considered as a fundamental state of proteins along with the structured native state and the solid-like amyloid state [15]. The signature of amyloid fibrils consists in pairs of closely mating β-sheets along the fibril axis; their presence is implicated in several degenerative pathologies triggered by aberrant protein mis-folding and subsequent aggregation [16].

On the other hand, there is much debate on the type of interactions that underlie protein LLPS at a molecular level. In fact, a central issue in the field is the ability of predicting which proteins can undergo LLPS in physiological conditions in living cells, based on the knowledge of the amino-acid sequence alone, in particular for IDRs [9]. The understanding of the sequence determinants of phase separation in IDRs is still basic, but it is clear that different flavors of IDRs exist that determine the type of stimulus the IDR responds to [17], depending as well on fluctuations in the microenvironment and on the specific context [18]. The sequence also likely determines the emergent properties of its dense phase, i.e., dense-phase concentration [19], and material properties such as visco-elasticity [20].

Multi-valent interactions of heterogeneous modular binding domains and their target motifs can drive LLPS of proteins with a well-defined native structure [21], yet the forces promoting LLPS of IDRs are less understood. A role has been suggested for several weak non-covalent interactions such as electrostatic, dipole-dipole, pi-pi stacking, cation-pi, hydrophobic and hydrogen bonding (namely β-zipper) interactions [4].

In particular, the importance of charge patterns has been highlighted [22], whereas pi-pi stacking interactions were found to involve non-aromatic as well as aromatic groups in folded globular proteins, so that a phase separation predictive algorithm was built based on pi interaction frequency [23].

The use of one or few of those features to predict protein LLPS is typical of the so-called first-generation phase separation predictors [24]. More recently, other approaches were proposed, in which either the multiple features associated with phase separation are comprehensively incorporated within a unique score [25] or LLPS prediction is based on the large conformational entropy associated with nonspecific side-chain interactions in the dense condensed state [26].

In this work, we wish to test the possible role played by β-strand pairing as one of the many interaction modes driving protein LLPS. In fact, the formation of β-sheets with a high degree of structural order is fundamental in both the structured native state (intra-chain pairing) and the amyloid state (inter-chain pairing), so that a subtle balance between different factors is at play [27].

On the one hand, we may expect a reduced β-pairing propensity for protein sequences driving LLPS. Accordingly, at the intra-chain level a disordered conformational ensemble would be favored for IDRs in the diluted state, whereas at the inter-chain level the droplet condensed state would be promoted over the amyloid state. On the other hand, an increase in β-pairing propensity may help the demixing of IDRs sequences.

We evaluated β-pairing propensity by means of the well-established PASTA algorithm [28,29], which allows to predict the presence of amyloid interactions between co-aggregating proteins [30]. The PASTA energy function evaluates the stability of putative cross-pairings between different sequence stretches and is based on knowledge-based statistical potentials, estimated separately for parallel and anti-parallel interactions.

We used different properties of the sequence stretches involved in the β-pairings with the best PASTA scores to build a scoring function which predicts the ability of a protein chain to drive LLPS. We also build a generalized score, by combining the PASTA properties with PScore, possibly the best performing first-generation predictor, based on pi-pi interactions [23]. Significant improvement over PScore in the performance of the generalized score would signal that β-pairing information may be crucial to better characterize LLPS behavior.

The development of effective LLPS predictors is very much depending on the availability of reliable LLPS datasets, for both positive [31,32,33] and negative [26] sets. The latter may be built based on proteomes from different organisms. Interestingly, our results depend on the choice of the positive set and, to a lesser extent on the choice of the negative set.

## 2. Materials and Methods

### 2.1. Data Sets

We considered two different positive sets of phase-separating proteins: the PP dataset, including 121 proteins, from the PhaSePro database (https://phasepro.elte.hu, accessed on 30 June 2020) [33], and the LLPS dataset, including 453 proteins, as compiled in [26] by merging different datasets. The LLPS set contains the PP sequences as a subset.

We as well considered two different negative sets, both compiled in [26]: the hsnLLPS set, including 18108 sequences from the Swiss-Prot human proteome after removal of human proteins included in phase separation datasets (such as LLPS), and the nsLLPS dataset, including 3911 proteins from different organisms (*C. elegans*, *C. reinhardtii*, *D. melanogaster*, *H. sapiens*, *M. musculus*, *R. norvegicus*, *S. cerevisiae*, *S. pombe*, *X. laevis*), reflecting the organism composition of the LLPS dataset.

The script provided in [23] does not output any PScore for several sequences from the LLPS, hsnLLPS, nsLLPS datasets compiled in [26]. This occurs when the sequence length is smaller than 140 residues (the length of the shortest sequence in the PP dataset used in [23] as a positive set) or when letters different from the ones coding for the 20 natural amino acids are present in the protein FASTA sequence.

Since our analysis is based on the use of PScore, we removed those sequences from the LLPS, hsnLLPS, nsLLPS datasets used in this work, which then contain 442, 16,360, 3503, sequences, respectively.

### 2.2. PASTA Score, β-Pairing Length, and Register Shift

For each of the protein sequences in the data sets we use the PASTA algorithm, introduced to predict whether a given sequence may stabilize the aggregated amyloid structure [28]. The algorithm proved useful to study the role of sequence heterogeneity in driving specific aggregation into ordered self-propagating cross-β structures [29]. The PASTA algorithm associates a score to all inter-chain β-pairings that can be built by hydrogen-bonding together two, possibly overlapping, sequence stretches with the same length.

In this approach one assumes that only a single sequence stretch per chain participates in the β-pairing and that all other residues are not involved in aggregation and are found in a disordered non-compact conformation. The PASTA score for a given β-pairing is based on pairwise sequence specific energy parameters, estimated to evaluate the propensity of two given residues to be found in front of each other within nearby β-strands. Two distinct sets of energy parameters are used for β-pairings with either a parallel or an anti-parallel orientation. We use the energy parameters presented in the PASTA2 update [30].

For each sequence in the positive and negative sets used in this work, we determine the 5 “best pairings”, that is characterized by the lowest PASTA scores, between two amino acid chains sharing the same sequence. They are the pairings more likely to engage into and stabilize a cross-β structure. In the work presented here, we use *E*, the PASTA score of the best pairing (that is, the lowest PASTA score), $l_{p}$, the length of the best pairing (that is, the number of residues in each sequence stretch involved in the pairing), *S*, the average register shift for the 5 best pairings. In-register pairing (that is, no register shift) would correspond to a sequence stretch paired with itself.

The register shift can be obtained using the indices of the amino acids flanking the stretches involved in the pairing, as shown in Figure 1. For the $\left[k,l\right]$ sequence stretch (k<l) paired with the $\left[m,n\right]$ sequence stretch (m<n, $n-m=l-k=l_{p}-1$), the register shift is defined as $S=\left|m-k\right|=\left|n-l\right|$ for both the parallel and the anti-parallel orientation. The average register shift over different best pairings is considered because the best pairing is almost always in-register [28].

### 2.3. PScore Computation

The scoring function PScore was introduced in [23] to predict liquid-liquid phase separation of a protein sequence. PScore is a composite scoring function obtained by linearly combining eight different terms. The main focus of PScore is highlighting the role of planar pi-pi interactions, involving also non-aromatic groups. We used the python code available at https://doi.org/10.7554/eLife.31486.021 to calculate the PScores used in this work. A detailed description of PScore and of its training can be found in [23]. As mentioned in Section 2.1, we could not obtain PScore for input sequences shorter than 140 residues or with unusual amino acid letters in their FASTA sequence.

### 2.4. Performance Evaluation

The scoring functions used in this work are used to predict whether a protein sequence is involved in liquid-liquid phase separation (score higher than a threshold value) or not (score lower than the threshold). Their performances are evaluated in terms of the area under the curve (AUC) in receiver operator characteristics (ROC) space, by calculating the number of true positives (TP), false negatives (FN), true negatives (TN), and false positives (FP).

As illustrated in Figure 2a, sequences in the positive set can be classified by the scoring function either correctly (as true positives) or incorrectly (as false negatives), such that their total number is P=TP+FN. Similarly, sequences in the negative set can by classified by the scoring function correctly (as true negatives) or incorrectly (as false positives), such that their total number is N=TN+FP. The ROC space is defined with false alarm rate (false positive rate FPR=FP/N, the fraction of sequences in the negative set incorrectly classified) along the horizontal axis and sensitivity (true positive rate TPR=TP/P, the fraction of sequences in the positive set incorrectly classified) along the vertical axis.

In the limit of very high (+∞) threshold, no sequence is predicted as positive, so that all negatives are predicted correctly yet no positive is and FPR=TPR=0. In the limit of very low (−∞) threshold, all sequences are predicted as positives, so that all positives are predicted correctly yet no negative is and FPR=TPR=1. The ROC curve joins these two points, using the (FPR, TPR) values obtained by changing the threshold all the way from +∞ to −∞. For an ideal classifier a range of threshold values exists such that both all positives and negatives are predicted correctly (TPR=1 while FPR=0), resulting in the whole space being under the ROC curve and therefore the maximum AUC=1. The larger the AUC, the better the classifier. A random classifier would on average increase TPR and FPR at the same rate, as the threshold is decreased, resulting in a straight ROC line with AUC = 0.5 (see Figure 2b for a pictorial illustration of AUC curves).

We also estimate the Matthews correlation coefficient (MCC), as another measure of the quality of the binary classification of protein sequences with respect to their phase separating behavior. MCC can be defined for a given point on the ROC curve as:(1)$$\mathrm{MCC}=\frac{\mathrm{TP}\times \mathrm{TN}-\mathrm{FP}\times \mathrm{FN}}{\sqrt{\left(\mathrm{TP}+\mathrm{FP})(\mathrm{TP}+\mathrm{FN})(\mathrm{TN}+\mathrm{FP})(\mathrm{TN}+\mathrm{FN}\right)}}\;.$$

One then considers the maximum MCC which is obtained by varying the threshold value used for prediction along the ROC curve. MCC=0 both in the bottom left (TP=FP=0) and in the top right (TN=FN=0) corner of the ROC curve. AUC=MCC=1 for an ideal binary classifier, whereas AUC=1/2, MCC=0 for a random classifier.

However, care should be taken in comparing MCC values obtained for data sets with different sizes. According to Equation (Equation 1), MCC→0 for N=TN+FP→∞ (the size of the negative set), with a fixed size of the positive set P=TP+FN and both false alarm rate FP/N and sensitivity TP/P fixed. Thus as a consequence of taking a larger negative set, the value of MCC on the test set decreases even if maintaining the performance for sensitivity and false alarm rate at a similar level.

### 2.5. Training Procedure by *k*-Fold Cross Validation

Cross validation is a very useful technique to assess the performance of any predictive machine learning algorithm. It is basically a resampling method that takes into account different parts of the full data as the test set and then the remaining part as the training set. For a *k*-fold cross validation algorithm, the entire dataset, that is the union of the positive and negative set, is randomly partitioned into *k* subsets of equal size. In our work, we choose one among the *k* subsets as the test set and the union of the remaining k−1 ones as the training set. During the training procedure for each of our defined scoring functions we optimised the weights (α, β or γ) by maximising AUC on the training set. In all the cases, the simplex algorithm was used to maximise the AUC on the training set [34].

Then using the optimised parameters we evaluate AUC (and MCC) on the test set. This process is repeated for *k* times, where in each attempt a different subset is in turn chosen as the test set. Thus, after each step we end up with *k* set of values for the optimised parameters, maximised AUC on the training set and estimated AUC (and MCC) on the test set. Then we repeat the random partitioning of the entire data set into *k* subsets for a total of *M* realisations. This gives us a total of k×M set of outputs, which we then use for the statistical analysis as shown in Figure 3, Appendix A. We used k=5 and M=25.

We additionally assess the average performance of PScore on the test sets used for the k-fold cross-validation procedure discussed above. Note that PScore parameters were already trained previously [23].

## 3. Results

In this contribution we consider a number of possible scoring functions where we include different features from the output of the PASTA algorithm [30], originally introduced to predict the propensity to aggregate into amyloid structure, and combine them with PScore, a phase-separation predictor built on the frequency of pi-pi contacts [23]. We then estimate their abilities to classify a protein sequence according to its phase separating behaviour. We have compared our results with respect to the original Pscore as well.

### 3.1. Data Sets

We considered two possible choices for both the positive set (the sequences that are known to undergo phase separation) and the negative set (the sequences that do not undergo phase separation) used to train the scoring functions.

In all cases, the protein sequences were taken from already published data sets (see Section 2.1).

#### 3.1.1. Positive Sets

As a first possibility for the positive set we use PP (https://phasepro.elte.hu, accessed on 30 June 2020) [33], which provides manually curated protein regions from a variety of organisms, whose association with liquid–liquid phase was experimentally validated in the literature, either “in vitro” or “in vivo”. PP was used as a positive set for the development and training of PScore [23].

As a second possibility for the positive set we use LLPS [26], obtained by merging PP with two other data sets for liquid-liquid phase separation, REV and LPS-D. The REV data set is a subset of PhaSepDB (http://db.phasep.pro/, accessed on 30 December 2020); it includes proteins whose involvement in either “in vivo” or “in vitro” liquid–liquid phase separation can be found in the literature [32]. The LPS-D data set is a subset of LLPSDB (http://bio-comp.org.cn/llpsdb, accessed on 30 December 2020); it collects “droplet-driving” proteins, observed to undergo “in vitro” liquid–liquid phase separation spontaneously as one component, with well-defined experimental conditions and phase diagrams. LLPS thus includes PP as a proper subset.

#### 3.1.2. Negative Sets

Both choices for the negative set were assembled in [26]. As a first possibility we use hsnLLPS, the Swiss-Prot human proteome from which all proteins that appear in any of the liquid–liquid phase separation data sets were removed. As a second possibility we use nsLLPS, a collection of proteins sampled from the proteomes of 9 different organisms in order to reproduce the frequencies with which sequences from different organisms appear in the LLPS dataset, after removal of all proteins that appear in any of the liquid–liquid phase separation data sets.

For LLPS, hsnLLPS, and nsLLPS we had to consider only the entry sequences yielding a PScore, resulting in 442, 16,360, 3503, entries, respectively (see Section 2.1).

### 3.2. Scoring Functions

In this study we report the performances in predicting the phase separation behaviour of protein sequences for a total of six scoring functions, which are defined as follows: (2)$$\begin{array}{ccc} s_{0} & = & \left(\frac{E}{N}\right)+\tilde{\beta }\ln\left(S+1\right)\;, \end{array}$$(3)$$\begin{array}{ccc} s_{00} & = & \left(\frac{E}{N}\right)+\tilde{\beta }\ln\left(S+1\right)+\tilde{\gamma }\ln\left(l_{p}\right)\;, \end{array}$$(4)$$\begin{array}{ccc} s_{1} & = & \alpha \left(\frac{E}{N}\right)+P\;, \end{array}$$(5)$$\begin{array}{ccc} s_{2} & = & \beta \ln(S+1)+P\;, \end{array}$$(6)$$\begin{array}{ccc} s_{3} & = & \alpha \left(\frac{E}{N}\right)+\beta \ln\left(S+1\right)+P\;, \end{array}$$(7)$$\begin{array}{ccc} s_{4} & = & \alpha \left(\frac{E}{N}\right)+\beta \ln\left(S+1\right)+\gamma \ln\left(l_{p}\right)+P\;, \end{array}$$
where *E* and *P* are the best PASTA score and the PScore, respectively; *N* is the number of amino acids in the protein sequence; *S* is the average register shift over the best five β-pairings and $l_{p}$ is the length of the best β-pairing. All possible pairings between two different, possibly overlapping, stretches from the same sequence are searched for (see Section 2.2 for details on the PASTA algorithm and its different outputs used here; see Section 2.3 for details on the PScore).

The PASTA “energy density” E/N is used having in mind the mean field Flory-Huggins approach [7]. In the PASTA algorithm one specific β-pairing is assumed to form between two sequence stretches in two different chains while all other chain portions remain disordered; the contribution of this specific interaction to the phase separation of the full length chains needs then to be normalised by the chain length.

We use the logarithm of quantities that are in their essence numbers of consecutive residues along the chain, such as $l_{p}$ or *S*, because in this way they are most naturally connected to entropies. For example, the entropy loss associated to constraint such as loop closure or anchoring of one/both ends, imposed on otherwise conformationally heterogeneous segments of length *m*, scales like lnm.

In particular, the leading contribution to the entropy loss in going from the diluted to the dense phase due to β-pairing is proportional to $l_{p}$. This is in principle already taken into account within the PASTA energy parametrization [28]. The subleading contribution would be proportional to $\ln\left(l_{p}\right)$, and can be interpreted as the entropy loss due to anchoring the two sequence portions through either one or both the two end pairs in the pairing, while leaving the remaining pairs still free. By decreasing $l_{p}$, the entropy of the dense droplet phase is increased with respect to the dilute phase.

We observe that a similar argument can be used to estimate the entropy loss, due to the anchoring to the β-paired sequence stretch of the two sequence portions flanking it from either ends. By assuming $l_{p}\ll N$, with *l* the position along the sequence of the paired stretch, we estimate the entropy loss as $\ln\left(l\right)+\ln\left(N-l\right)$. This expression is maximum for l=N/2, so that the entropy of the dense droplet phase is higher when the β-paired sequence stretch is closer to one end of the chain than to its center.

Also, the entropy of the dense phase is increasing in the case of an off-register pairing because of two reasons, if we assume that only one β-pairing per chain is formed. First, two chains can pair two different sequence segments (say *A* and *B*) in two different ways (say *A* from chain 1 with *B* from chain 2 or *B* from chain 1 with *A* from chain 2), whereas for in-register pairing there is only one possibility, that is *A* from chain 1 with *A* from chain 2. In the case of a multi-chain condensate the combinatorics of the different possible arrangements may lead easily to a high entropy gain. Second, if we assume that in the amyloid phase both possible out-of-register pairings described above are formed, a constraint is then placed on the sequence segment between the two portions *A* and *B*, being $S-l_{p}+1$ residues long, implying an entropy loss that scales as $\ln\left(S+1-l_{p}\right)∼\ln\left(S+1\right)$, if $l_{p}\ll S$. The larger *S*, the higher the entropy difference favoring the condensed droplet state over the amyloid one.

### 3.3. The LLPS Positive Set

We optimised the parameters α, β and γ in the different scoring functions by maximising the AUC on the training set and then used the optimised parameters to find AUC and MCC on the test sets (see Section 2.4 for details on AUC and MCC). Training was performed using a *k*-fold cross validation with k=4 such that the full dataset (the union of the positive and the negative set) is split randomly in *k* equally sized subsets with one of them used in turn as the test set and the union of the other k−1 ones as the training set (see Section 2.5 for details on the cross validation training procedure).

The PScore function was already trained previously [23], so we evaluated its performance on the test set by computing the corresponding AUC and MCC for each realisation of *k*-fold cross validation, whereas we did not evaluate its performance on the training set.

In this section we show the results we have obtained by using LLPS as the positive set.

#### 3.3.1. Performance Evaluation: AUC on the Test Set

We begin by presenting in Figure 3 the results obtained with hsnLLPS as the negative set. In Figure 3a we show the normalised distributions of AUC on the training set. While $s_{0}$ and $s_{00}$ perform the worst, the performances of $s_{2}$, $s_{1}$, $s_{3}$, $s_{4}$ get increasingly better.

In Figure 3b we show the normalised distributions of AUC on the test set using the parameters optimised on the training set. The performances of $s_{0}$ and $s_{00}$ are worse than that for PScore, whereas $s_{2}$ performs similarly to PScore and $s_{1}$, $s_{3}$, $s_{4}$ perform increasingly better than PScore. All the trends observed for AUC distributions on training and test sets with hsnLLPS as the negative set, remains qualitatively the same if nsLLPS is instead chosen as the negative set (see Appendix A).

To check statistical significance we perform one-way ANOVAs on AUC with the scoring functions as the factors, using different combinations of them. The results of these ANOVAs are summarised in Table 1 for hsnLLPS as the negative set and in Appendix A for nsLLPS as the negative set.

In both cases, we found that the AUC values of the scoring function pairs $\left(P,s_{2}\right)$, $\left(s_{1},s_{3}\right)$, do not differ in a statistically significant way from each other, whereas the AUC values for other combinations of scores, such as $\left(P,s_{1},s_{2}\right)$ or $\left(s_{1},s_{3},s_{4}\right)$, are instead different in a statistically significant manner. Taken together, this shows that the terms E/N and $\ln\left(l_{p}\right)$ can be fruitfully added to PScore, in order to improve the performance of the scoring function, whereas the addition of the term $\ln\left(S+1\right)$ does not provide a statistically significant improvement.

A series of Kolmogorov-Smirnov tests on each of the AUC distributions on the test set for different scores (see Appendix A) found that the distributions are normal for all scores with both choices for the negative set (p>0.3) except for the score $s_{0}$ with hsnLLPS as the negative set (p=0.034).

#### 3.3.2. Performance Evaluation: MCC on the Test Set

The values of AUC and MCC on the test set evaluated using the parameters optimised on the training set indicate the performance of each of the scores to classify protein sequences according to their phase separating behaviour. In Table 2, we summarise the mean values of AUC and MCC on the test set for each of the scores, along with the optimised parameters as defined in Equations (2)–(6), with hsnLLPS as the negative set, whereas in Appendix A we show the same values obtained with nsLLPS as the negative set.

Although trends in the comparison between different scoring functions are similar for both negative sets, as we will discuss below, there is a clear quantitative difference between the two choices. Prediction against hsnLLPS is harder than against nsLLPS, resulting in lower AUC values.

For hsnLLPS as the negative set, $s_{4}$ is identified as the best performing scores with AUC ≈0.79, with a statistically significant improvement over PScore (AUC ≈0.75). Performances of the scores $s_{1}$ and $s_{3}$ are similar to each other and better than PScore, as also pointed out by the one-way ANOVA. Similar trends are obtained with nsLLPS as the negative set, with AUC ≈0.84 for PScore, improved to AUC ≈0.88 with score $s_{4}$.

Using hsnLLPS as the negative set, a weak improvement over the PScore performance in MCC brought about by the addition of the PASTA terms is apparent only for score $s_{4}$, whereas scores $s_{1}$ and $s_{3}$ improve AUC but not MCC with respect to PScore.

This can be rationalized by looking at the ROC curves shown in Figure 4a for each of the considered scoring functions. ROC curves are derived on the entire data set (LLPS as the positive set and hsnLLPS as the negative set), using the optimised parameters, which are summarised in Table 2. Composite scores obtained by combining PScore with PASTA terms improve AUC in the high sensitivity, low specificity portion of the ROC curve, whereas MCC is typically related to its low sensitivity, high specificity portion.

Similar trends are observed when nsLLPS is used as the negative set, with MCC =0.56 for score $s_{4}$ improving with respect to MCC =0.51 for PScore (see Appendix A and Figure 4c). The values of MCC are much higher for nsLLPS because of its much smaller size with respect to hsnLLPS (see Section 2.4).

#### 3.3.3. Optimised Weights

Finally, we have also studied the distribution of the weights used in the definition of the scoring functions. These parameters are optimised in the training set to produce the maximum AUC. Normalised distributions of the optimised parameters are shown in the first row of Appendix A for hnsLLPS as the negative set. From Kolmogorov-Smirnov tests (see Appendix A) we found that the obtained data follow a normal distribution in most cases (p>0.1), with α for scores $s_{1}$ and $s_{3}$ providing borderline cases (p≈0.02), and the only clear exception of $\tilde{\beta }$ in score $s_{0}$ with ($p\approx 3\cdot 10^{-14}$).

We note that the distributions of the α parameter for scores $s_{1}$ and $s_{3}$ are essentially the same (see Appendix A) and that the support of the distributions for the β parameter contains or is close to the 0 value (Appendix A). Both facts confirm that the contribution of the $\ln\left(S+1\right)$ term is not significant.

Similar trends are observed if nsLLPS is used as the negative set (see the third row of Appendix A), although in this case most distributions turn out to be not normal according to the Kolmogorov-Smirnov test (see Appendix A: the only parameters with a clearly normal distribution, p>0.05, are β in score $s_{3}$ and α and γ in score $s_{4}$). Despite this, the values and distributions of $\tilde{\gamma }$, α and γ obtained for the two considered negative sets are roughly consistent with each other, with the α parameter being scaled down by a factor of roughly 2 when considering nsLLPS in place of hsnLLPS as the negative set.

### 3.4. The PP Positive Set

In this section we show the results we have obtained by using PP as the positive set. The PScore function was already trained with PP as the positive set, and with a data set from the human proteome as the negative set [23], presumably similar to hsnLLPS.

#### 3.4.1. Performance Evaluation: AUC on the Test Set

We begin by presenting in Appendix A the results obtained with hsnLLPS as the negative set. In Appendix A we show the normalised distributions of AUC on the training set. While $s_{0}$ and $s_{00}$ perform the worst, and in a similar way, between each other, the performances of $s_{1}$, $s_{2}$, $s_{3}$, $s_{4}$ get increasingly better, although $s_{3}$ and $s_{4}$ show a quite similar distribution. In Appendix A we show the normalised distributions of AUC on the test set using the parameters optimised on the training set. The performances of $s_{0}$ and $s_{00}$ are similar to each other and worse than that for PScore, whereas $s_{1}$ appears to perform slightly better than PScore and $s_{2}$, $s_{3}$, $s_{4}$ perform similarly to each other and clearly better than PScore.

All the trends observed for AUC distributions on training and test sets with hsnLLPS as the negative set, remains qualitatively the same if nsLLPS is instead chosen as the negative set (see Appendix A).

To check statistical significance we perform one-way ANOVAs on AUC with the scoring functions as the factors, using different combinations of them. The results of these ANOVAs are summarised in Table 3 and Appendix A for both possible choices of the negative set.

In both cases, we found that the AUC values of the scoring function pair $\left(s_{0},s_{00}\right)$ and of the ones within the set $\left(s_{2},s_{3},s_{4}\right)$ do not differ in a statistically significant way from each other. On the other hand, the AUC values for other combinations of scores, such as $\left(P,s_{1}\right)$ or $\left(s_{1},s_{3},s_{4}\right)$, are instead different in a statistically significant manner. Taken together, this shows that the terms E/N and $\ln\left(S+1\right)$ can, each in its turn, be fruitfully added to PScore, in order to improve the performance of the scoring function. The resulting improvement, however, is much better for the $\ln\left(S+1\right)$ term, so that the further addition of the E/N term (i.e., going from score $s_{2}$ to score $s_{3}$) does not provide a statistically significant improvement. On the other hand, the addition of the term $\ln\left(l_{p}\right)$ does not provide a statistically significant improvement under any condition.

A series of Kolmogorov-Smirnov tests on each of the AUC distributions on the test set for different scores (see Appendix A) found that the distributions are normal for all scores with both choices of the negative set (p>0.1).

#### 3.4.2. Performance Evaluation: MCC on the Test Set

In Table 4, we summarise the mean values of AUC and MCC on the test set for each of the scores, along with the optimised parameters as defined in Equations (1)–(5), with hsnLLPS as the negative set, whereas in Appendix A we show the same values obtained with nsLLPS as the negative set.

Although trends in the comparison between different scoring functions are basically similar for both negative sets, as we will discuss below, there is a clear quantitative difference between the two choices. Prediction against hsnLLPS is harder than against nsLLPS, resulting in lower AUC values.

For hsnLLPS as the negative set, any of the scores $s_{2},s_{3},s_{4}$ can be identified as the best performing one, with AUC ≈0.83, with a statistically significant improvement over PScore (AUC ≈0.78). The performance of the score $s_{1}$ (AUC ≈0.80) is also better than PScore, as also pointed out by the one-way ANOVA. Similar trends are obtained with nsLLPS as the negative set (see Appendix A), with AUC ≈0.87 for PScore, improved to AUC ≈0.92 with scores $s_{3},s_{4}$.

Using hsnLLPS as the negative set, a weak improvement over the PScore performance in MCC brought about by the addition of the PASTA terms is apparent only for score $s_{4}$ (0.26 vs. 0.25), whereas scores $s_{2}$ and $s_{3}$ have the same AUC as score $s_{4}$, but do not improve MCC with respect to PScore. Score $s_{1}$ has a higher AUC, yet the same MCC as Pscore.

This can be rationalized by looking at the ROC curves shown in Figure 4b for each of the considered scoring functions. ROC curves are derived on the entire data set (PP as the positive set and hsnLLPS as the negative set), using the optimised parameters, which are summarised in Table 4. Composite scores obtained by combining PScore with PASTA terms improve AUC in the high sensitivity, low specificity portion of the ROC curve, whereas MCC is typically related to its low sensitivity, high specificity portion.

Similar trends are observed when nsLLPS is used as the negative set, with MCC =0.50 for score $s_{4}$ improving with respect to MCC =0.46 for PScore (see Table 4 and Figure 4d). The values of MCC are much higher for nsLLPS because of its much smaller size with respect to hsnLLPS (see Section 2.4).

#### 3.4.3. Optimised Weights

Finally, we have also studied the distribution of the weights used in the definition of the composite scoring functions. These parameters are optimised in the training set to produce the maximum AUC. Normalised distributions of the optimised parameters are shown in the second row of Appendix A for hnsLLPS as the negative set. From Kolmogorov-Smirnov tests (see Appendix A) we found that the obtained data follow a clearly normal distribution only for all parameters (α, β, γ) in score $s_{4}$ (p>0.1), whereas all other cases are borderline (0.01<p<0.05), with the exception of $\tilde{\beta }$ for score $s_{0}$ which is clearly not normal ($p\approx 10^{-16}$).

We observe that the distributions of the the β parameter for scores $s_{2}$, $s_{3}$, $s_{4}$, are very similar to each other (Appendix A) and that the support of the distribution for the γ parameter contains the 0 value (Appendix A). These facts confirm that the contribution of the $\ln\left(l_{p}\right)$ term is not significant and the contribution of the $\ln\left(S+1\right)$ term is the dominant one.

Similar trends are observed if nsLLPS is used as the negative set (see the fourth row of Appendix A), with the exception of the distributions of the β parameter, which become bimodal ones. With nsLLPS as the negative set, most distributions turn out to be clearly not normal ($p<2\cdot 10^{-3}$) according to the Kolmogorov-Smirnov test (see Appendix A), with the exception of α in scores $s_{1}$, $s_{3}$, $s_{4}$, and γ in score $s_{4}$ (p>0.05).

Overall, the values and distributions of all parameters obtained for the two considered negative sets are roughly consistent with each other. Interestingly, the α parameter is scaled down by a factor of roughly 2 when considering nsLLPS in place of hsnLLPS as the negative set, with PP as the positive set as well.

### 3.5. Sequences Classified Differently by $s_{4}$ and PScore

In order to better characterize why the addition of the PASTA related terms to the PScore allows to improve the classification of phase separating proteins, we selected in LLPS the 44 protein sequences, set ${\tilde{S}}_{4}$, which are correctly classified as positives by the score $s_{4}$ but they are not by PScore, against the negative set hsnLLPS, if a false positive rate FPR=0.3 is considered as the precision threshold. We chose FPR=0.3 because it is the part of the ROC curve where the improvement brought about by $s_{4}$ over PScore is the clearest, with 319 true positives detected by score $s_{4}$, set $S_{4}$, against 296 detected by PScore, set $S_{P}$. Conversely, we also selected in LLPS the 21 sequences, set ${\tilde{S}}_{P}$, that are correctly classified by PScore but not by $s_{4}$, against the negative set hsnLLPS, at FPR=0.3. We restrict this analysis for clarity to the larger positive set and the larger negative set.

The different features in the sequences in the two sets should encapsulate the biophysical properties captured by the PASTA related terms. Analyzing them we observe some common trends. Low-complexity domains are present to some extent in both sets (a more detailed study of the intrinsic disorder content is presented in the next Section 3.6). Yet we observe that FUS and the other low-complexity domains studied in [35] or proteins enriched in proline/glycine residues that self-organize into elastomeric assemblies [36] are not present in both sets. In fact, the sequences mentioned above are classified correctly already by PScore alone (and remain correctly classified by $s_{4}$). In fact, the energy parameters used in PASTA typically favors the β-pairing of hydrophobic residues, making the algorithm less suitable to investigate β-pairing in low-complexity prion-like domains, for which “ad hoc” predictors are typically developed [37]. The high propensity for phase separation of low-complexity domain is at any rate well captured by PScore, and maintained upon addition of the PASTA-related terms.

We instead observe in set ${\tilde{S}}_{4}$ the presence of generally short (4-8 residues) hydrophobic stretches, the ones picked up by the PASTA algorithm for the best β-pairing (always a parallel in-register pairing), enriched in V,I,L,F residues. In set ${\tilde{S}}_{P}$, the stretches selected by PASTA become longer than in set ${\tilde{S}}_{4}$. This is not surprising since the score $s_{4}$, optimised for LLPS as the positive set against the negative set hsnLLPS, penalizes the increase in $l_{p}$, the length of the stretch. For 3 sequences in set ${\tilde{S}}_{P}$, the best β-pairing takes place with an in-register anti-parallel arrangement, wheres all other pairings are in-register parallel as in set ${\tilde{S}}_{4}$. Interestingly, most of the stretches are flanked by several charged/polar residues in both sets. We provide a list of the sequence stretches selected by PASTA along with the corresponding sequences in the Appendix A for both sets ${\tilde{S}}_{4}$ and ${\tilde{S}}_{P}$.

A non trivial difference between the two sets can be detected if we consider the position within the chain of the sequence stretch involved in the best β-pairing predicted by PASTA. Interestingly, as discussed in Section 3.2, this feature may affect the entropy of the droplet state. If *m* is the position of the initial residue in the stretch, *n* the position of its final residue, and *N* the overall length of the sequence, we can compute the fractional position *f* of the stretch along the sequence as
(8)$$f=\min\left\{\frac{m+n}{2N},1-\frac{m+n}{2N}\right\}\;.$$

With this definition, 0<f≤1/2: the closer the stretch is located to the center of the sequence, the higher the value of *f*. If we compute the average fractional position *f* of the sequence stretches selected by PASTA for the best β-pairing, we obtain f=0.232±0.020 for set ${\tilde{S}}_{4}$ and f=0.158±0.020 for set ${\tilde{S}}_{P}$. The fractional positions of the stretches in the two sets differ in a statistically significant way with a *Z*-score Z=3.7, with the stretches in set ${\tilde{S}}_{P}$ located closer to the chain ends.

As discussed in Section 3.2, the sequences in set ${\tilde{S}}_{P}$ would then increase the entropy of the droplet state with respect to the sequences in set ${\tilde{S}}_{4}$. Not capturing this effect within the $s_{4}$ scoring function might explain its failure in classifying correctly the sequences in set ${\tilde{S}}_{P}$, providing a clue for further improvement of LLPS prediction.

### 3.6. Intrinsic Disorder Prediction for Different Sequence Sets

In order to gain a further insight into the biophysical interpretation of our results, we computed the intrinsic disorder scores of different sequence sets using the MobiDB-lite consensus predictor, which combines a set of eight complementary intrinsic disorder predictors [38]. The results for the total disorder fraction (the fraction of residues classified as disordered) and the mean length of the predicted disordered segments are summarized in Table 5.

The two negative sets differ with respect to their intrinsic disorder content, with hsnLLPS being more disordered than nsLLPS. This is expected, since the human proteome is known to be characterized by a higher fraction of intrinsic disorder with respect to the proteomes of less complex organisms [39], such as the unicellular ones that are sampled to build nsLLPS (see Section 2.1).

The positive set LLPS has a much higher intrinsic disorder content than both negative sets. This is as well expected since both negative sets include a fraction of non disordered proteins with a well defined native structure, whereas the phase-separating proteins collected in LLPS are characterized by IDRs.

We observe that the fraction of intrinsic disorder and the mean length of disordered segments are similar in the sets $S_{4}$ and $S_{P}$, the true positives predicted by the score $s_{4}$ and PScore in LLPS with a decision threshold FPR=0.3. Both quantities are higher than in the full LLPS set, suggesting that both scores are better in detecting phase-separating sequences with a higher disorder content.

More interestingly, the increase in performance by $s_{4}$ over PScore is due to the ability in rescuing the correct prediction of phase-separating sequences (the ones in set ${\tilde{S}}_{4}$) with an average disorder content even lower than the full LLPS set and actually on par with the whole negative set hsnLLS from the human proteome. On the other hand, this is achieved at the price of losing the correct prediction of phase-separating sequences (the ones in set ${\tilde{S}}_{P}$) with a disorder content higher than for set ${\tilde{S}}_{4}$), yet still lower than for the full LLPS set.

The trends across the different sequence sets shown in Table 5 are also found by the standalone predictors considered in MobiDB-lite to calculate consensus. The intrinsic disorder scores predicted by some of them, Espritz [40], IUPred [41], GlobPlot [42], are shown in Appendix A.

Taken together, these results show that PScore is very effective in predicting phase-separating sequences with a high content of intrinsic disorder, whereas the contribution from the PASTA related terms is most effective for phase-separating sequences with a lower content of intrinsic disorder.

### 3.7. Protection of the Droplet State from the Amyloid State: A Hypothetical Scenario

The results presented so far show that the correct discrimination of phase-separating sequences can be improved by complementing PScore with terms related to the β-pairing between hydrophobic stretches. Rewarding the increase in the main chain conformational entropy associated to the β-pairing (see Section 3.2) is one of the factors allowing an improved performance over PScore. We hypothesize that the entropic stabilization of the droplet dense phase may be important against the amyloid state.

To test this conjecture, we plot the score $s_{4}$ for all sequences in the positive set against the PASTA energy *E*, used in this test as a predictor of the amyloid state. The lower (the more negative) *E*, the more likely the amyloid state according to the PASTA algorithm. As shown in Figure 5, the PASTA energy *E* displays a mild positive correlation with the score $s_{4}$ for both positive sets. The higher the value of the score $s_{4}$, the less likely the amyloid state, consistently with the idea that score $s_{4}$ is connected, at least in part, to the stabilization of the droplet state against the amyloid state.

Overall, we believe that the results obtained in this work fit consistently within the following general picture. The physico-chemical forces driving structure formation are the same in all protein states, including native folds in the diluted phase, the droplet and the amyloid state; the most stable state is almost inevitably the amyloid one and the droplet state is in general conducive towards it [43]. On this general ground, intrinsic disorder can be seen as protective against amyloid formation, as the examples of low-complexity domains and of proline/glycine-rich sequences discussed in Section 3.5 indeed show. In fact, amyloid state predictors can be built as the “reverse” of disorder predictors [44].

PScore classifies correctly phase-separating sequences with an intrinsic disorder higher than the average one in the LLPS positive set (see Table 5, set $S_{P}$). Those sequences, therefore, would be well protected against amyloid formation, even in the droplet state, although single point mutations can dramatically shift this balance [45]. The PASTA related terms helps in rescuing the prediction of precisely those few sequences that would be in need of a greater protection against the amyloid state, with an intrinsic disorder much lower than the average one in the positive set (see Table 5, set ${\tilde{S}}_{4}$).

The signal we uncover is small yet significant and is consistent with the hypothesis that the greater protection against the amyloid state needed by such sequences is provided by decreasing both the PASTA energy density and the length of the best β-pairing. Attaining the droplet state would be too dangerous for sequences with that disorder content and with a lower energy density and/or a longer β-pairing.

The PASTA-complemented scoring function then backfires for some sequences with a disorder content again lower than the average one in the full positive set (see Table 5, set ${\tilde{S}}_{P}$), classifying them as too dangerous for the droplet state. We can then hypothesize that other protection factors should be included into the scoring function or/and that the balance between the protection from the amyloid state provided by short β-pairings and the one provided by intrinsic disorder is not correctly described by the scoring function $s_{4}$.

A clue towards a possible extension of the scoring function is given by the observation (see Section 3.5) that in the backfire group (set ${\tilde{S}}_{P}$) the stretch involved in the best PASTA pairing is located further from the center of the chain with respect to the sequences in the rescued group (set ${\tilde{S}}_{4}$). This may suggest that the presence of a longer disordered sequence portion, from the β-paired stretch to the end of the chain, could be more protective against the amyloid state. This hypothesis is in fact consistent with the estimation of how the entropy of the β-pairing depends on the length of the two sequence portions flanking the β-paired stretch (see Section 3.2). The entropy is higher when the latter is located closer to the chain ends, thus suggesting that incorporation of this entropy term in the scoring function $s_{4}$ could further improve the prediction of phase-separating sequences.

## 4. Discussion

The formation of droplets of bio-molecular condensates through liquid-liquid phase separation of their component proteins is now recognized as a key factor in the maintenance of cellular homeostasis [46]. Therefore, the molecular principles that drive protein LLPS are under intense investigation.

On the one hand, this allowed to identify different protein properties, which are important in LPPS onset [7]. One or a few of these properties are at the basis of the so-called first-generation predictors [24], which try to discriminate LLPS proteins against the rest of the proteome. One notable example is PScore [23], a LLPS predictor based on a composite scoring function obtained by linearly combining eight different terms. The main focus of PScore is highlighting the role of planar pi-pi interactions, involving also non-aromatic groups. Very recently, a second-generation predictor of IDR-driven phase separation, combining a broad set of different physical interactions, was put forward [47].

On the other hand, however, the fact the interactions driving LLPS are redundant and multivalent has been recognized as a crucial aspect [48], leading to the suggestion that LLPS is stabilized by the large conformational entropy associated with non specific side-chain interactions and to the development of a related LLPS predictor [26].

Another important aspect is the recognition that some proteins can undergo LLPS by themselves and form single component condensates, whereas other proteins can be involved in LLPS, either to be recruited into or to form multi-component condensates, only in the presence of partner proteins. LLPS predictors can be developed separately for the two protein classes [49].

In this work we pursued a minimal approach, building a phase separation predictor which combines two specific and seemingly unrelated protein features, such as the ability to form pi-pi interactions and the propensity to stabilize a cross-β structure. The latter is formed in the amyloid aggregates, which are involved in degenerative diseases [16]. Biomolecular condensates may eventually solidify into an amyloid-like state [15], and disease-related mutations exist, which are known to accelerate this process [43,50].

How protein condensates can be protected from transitioning into the amyloid state is the subject of several studies. For example, while it was proposed that certain low complexity domains may adopt cross-β interactions that are structurally specific, they are also labile to disassembly, critically for protection from amyloid, leading to the formation of biomolecular hydrogels, as first recognized in [35], which can be described as dynamic non pathogenic fibers. Such labile cross-β interactions were suggested to be crucial for the assembly of protein condensates in low complexity domains, although the issue is a debated one [45]. The protecting role of proline and glycine residues, allowing protein condensates to remain amorphous without forming the extended β-structure conducive to the amyloid state, was instead investigated in the context of elastomeric assemblies [36]. The role of different water hydrogen bond arrangements in phase separation was also recently highlighted [51].

In practice, we studied the performance in predicting LLPS behaviour of few scoring functions (see Equations (Equation 2)–(Equation 7)) obtained by combining in different ways PScore, based on pi-pi interactions, with several outputs from the PASTA algorithm [29,30], which was introduced to predict the stabilization of amyloid structure. In particular, the PASTA algorithm here ranks all possible β-pairings between two portions from the same sequence (see Figure 1) according to their PASTA energy (the lower the energy the more stable the pairing). The β-pairing is assumed to take place between nearby β-strands through main chain hydrogen bonds established between different chains. In the context of protein-protein complexes, β-strand addition is a well known interaction mode. The PASTA algorithm does not consider, for simplicity, intra-chain β-pairings. This is justified ‘a posteriori’ by the finding that the lowest energy pairings are almost always in register [28], as also found for the sequences analyzed in this work (see Appendix A for a sequence subset).

The PASTA outputs that we tried to combine with PScore were the PASTA energy density (the lowest β-pairing energy normalised by the length of the protein sequence), the length of the lowest energy pairing and the average register shift (see Figure 1) of the five lowest energy pairings. We considered the logarithm of the pairing length and of the average register shift because they can be connected to the main chain conformational entropy of the condensate, as explained in Section 3.2. We observe that the PScore and the PASTA energy used here are in fact effective free energies which, in principle, take into account the presence of solvent molecules and the entropic nature of the hydrohobic effect in an implicit manner. Studying how phase separation may depend on temperature, for example distinguishing between sequences with either upper or lower critical solution temperature [52,53], is beyond the scope of this work.

We studied the dependence of our results on considering two different positive sets, PP and LLPS, of proteins known to be involved in liquid-liquid phase separation. PP was used for the training of PScore, whereas LLPS is a larger set, enriched in “droplet-driving” proteins with respect to PP, including it as proper subset. We as well studied the dependence of our results on considering two different negative sets, against which phase-separating proteins should be recognized. We considered hsnLLPS, essentially the human proteome from which proteins in the positive sets were removed [26], and nsLLPS, a mixture of proteins not involved in liquid-liquid phase separation from different organisms [26].

Our results show that the performance of PScore can indeed be improved in a statistically significant manner, as evaluated by AUC under the ROC curve, by adding to it contributions from the PASTA outputs (see Table 1). This holds to similar extents for both choices of the positive set (see Figure 4), once we take into account that LLPS is a slightly harder set to predict than PP for all scoring functions tested in this work. The possibility of improving the performance of phase-behavior predictor by combining two distinct unrelated features is a confirmation that several interaction modes can drive protein phase separation.

On the other hand, the general performance is greatly affected by the choice of the negative set. Discrimination against the human proteome turned out to be much harder, for all scoring functions tested in this work, than against the negative set built to represent proteomes from different organisms. In fact, the amount of intrinsic disorder in a proteome is known to increase with organism complexity [39]. Since 3 from the 9 proteomes represented in nsLLPS are from unicellar organisms (1 alga and 2 yeasts), this explains the above observation, as explicitly verified by the intrinsic disorder predictions reported in Table 5, because it is harder to discriminate phase-separating proteins against a negative set enriched with intrinsic disorder [23].

In this view, we can interpret the differences between the distributions of the optimised parameters which are observed when changing the negative set, in particular the decrease of the optimised α parameter when nsLLPS is used as the negative set (compare for example the first and the third row of Appendix A). We hypothesize that the PASTA energy density (the term in the scoring function weighted by α) may have a role in helping PScore to discriminate phase-separating sequences against other IDRs; the higher the intrinsic disorder content (for hsnLLPS), the higher the value of α needed for improvement over PScore.

We discuss now how our results depend on the different positive sets used in this work. Interestingly, improvement with respect to PScore is achieved in different ways depending on the choice of the positive set. In this respect, the effect of the choice of the negative set was instead a minor one (see also [54], where the PP positive set was used together with a negative set based on the human proteome, yet different from hsnLLPS).

When using LLPS, the larger positive set, enriched in “droplet-driving” proteins, both the PASTA energy density and the length of the lowest energy pairings were instrumental in improving the AUC performance of PScore in a statistically significant manner (see Table 3 and Figure 4). Since the corresponding weight α is positive (see Table 2), the higher its PASTA energy density, the more likely a protein chain will phase separate. Note that, since the β-pairing with the lowest PASTA energy always has a negative energy, a higher (i.e., less negative) energy density can be achieved by increasing (i.e., making less negative) the lowest PASTA energy for a given chain length, or by increasing the chain length for a given PASTA energy.

On the other hand, since the corresponding weight γ is negative (see Table 2), the lower the length of its lowest energy β-pairing, the more likely a protein chain will separate. Taken together, all the above conclusions are consistent with the notion that phase-separating sequences favour weaker and shorter β-pairings, thereby increasing β-structure lability, in order to stabilize entropically the droplet state against the more ordered amyloid state [15]. Within the PASTA algorithm descriptors, this can be done by either increasing the PASTA energy or by decreasing the length of β-pairing or by increasing the chain length for a given PASTA energy.

Instead, when using PP, the smaller positive set, only the average register shift significantly contributed to the improvement of the PScore (see Table 3 and Figure 4). Since the corresponding weight β is positive (see Table 4), the higher its average register shift, the more likely a protein chain will phase separate. This can be interpreted again as a way to stabilize entropically the droplet state against the amyloid state, by allowing for more heterogeneous β-pairing possibilities (see Section 3.2 for details).

In all cases, the improvement over the performance of PScore brought about by using the PASTA algorithm descriptors, is most effective in the high sensitivity, low specificity part of the ROC curve (see Figure 4). As a result, the Matthews Correlation Coefficient did not improve much with respect to PScore (see Table 2 and Table 4).

The analysis of the sequences (in the LLPS positive set) for which the improvement over PScore is effective reveals that they are characterized by the presence of short hydrophobic stretches, detected by PASTA for the β-pairing (see Appendix A). Most importantly, the intrinsic disorder content predicted by a consensus method clearly shows that those sequences have a much lower disorder content than the average one in LPPS, or the one of the sequences correctly classified by PScore (see Table 5 and Appendix A). This could be interpreted in terms of the protection needed by those sequences against the amyloid state, a protection possibly provided by the increase of the β-pairing entropy in the droplet state. Along the same lines, a possible extension of the scoring functions used in this work can be envisioned (see Section 3.7).

To conclude, our analysis consistently suggests that, when the β-pairing interaction mode is considered as a determinant of LLPS together with pi-pi interactions, protein sequences are more likely to be involved in phase separation if the main chain conformational entropy of β-pairing maintained droplet state is increased. This would stabilize the droplet state against the more ordered amyloid state. Interestingly, the entropic stabilization of the droplet state appears to proceed according to different mechanisms, depending on the fraction of “droplet-driving” proteins present in the positive set. This latter finding is definitely worth further investigation, for example by refining the positive set composition and/or by studying different entropic stabilization mechanisms in binding mode interactions other than β-pairing.

## Figures and Tables

**Figure 1 biomolecules-12-01771-f001:**
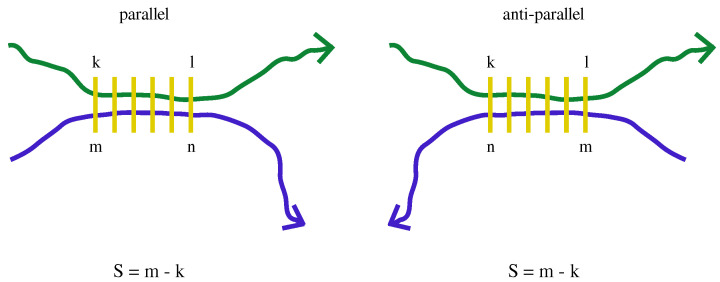
Schematic diagram showing β-pairings between two amino acid chains for parallel and anti-parallel orientations. The pairing amino acids are connected by yellow lines. The register shifts in both cases are indicated below the diagrams. In this work the two chains share the same sequence.

**Figure 2 biomolecules-12-01771-f002:**
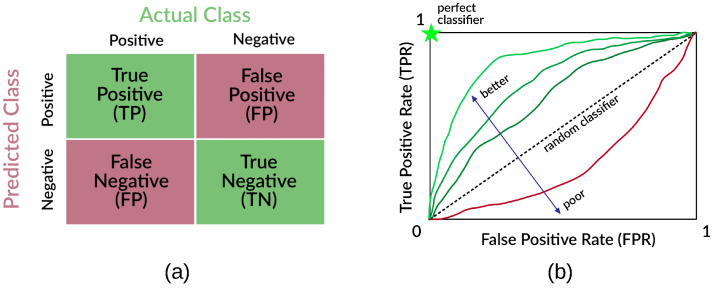
Pictorial illustration of (**a**) sequence partition according to classifier outcome; (**b**) different examples of ROC curves.

**Figure 3 biomolecules-12-01771-f003:**
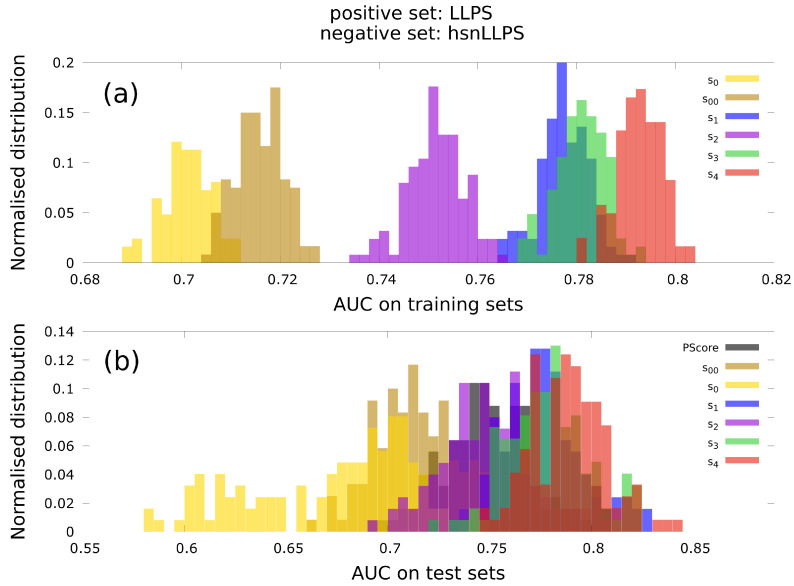
Normalised distributions for the values of AUC on the (**a**) training set and the (**b**) test set as obtained from the cross validation procedure. LLPS is the positive set and hsnLLPS is the negative set. The AUC values for PScore shown here were obtained on the test sets of cross validation.

**Figure 4 biomolecules-12-01771-f004:**
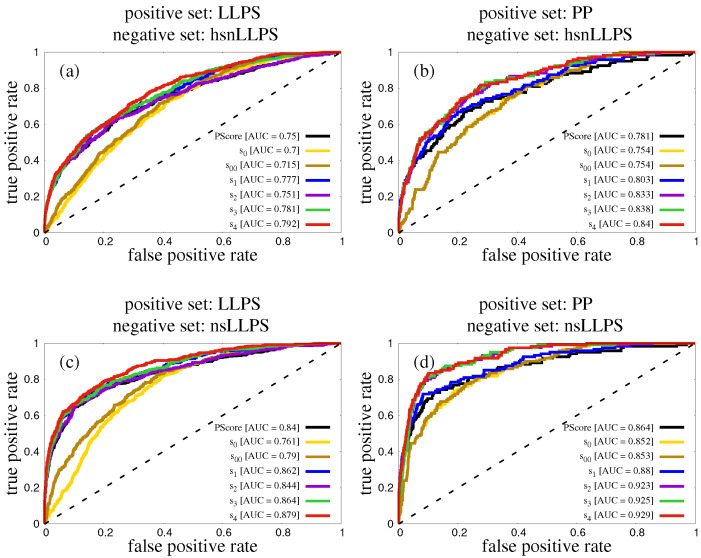
ROC curves corresponding to different predicting scores. The dashed line represents the random classifier. (**a**) LLPS positive set, hsnLLPS negative set. (**b**) PP positive set, hsnLLPS negative set. (**c**) LLPS positive set, nsLLPS negative set. (**d**) PP positive set, nsLLPS negative set.

**Figure 5 biomolecules-12-01771-f005:**
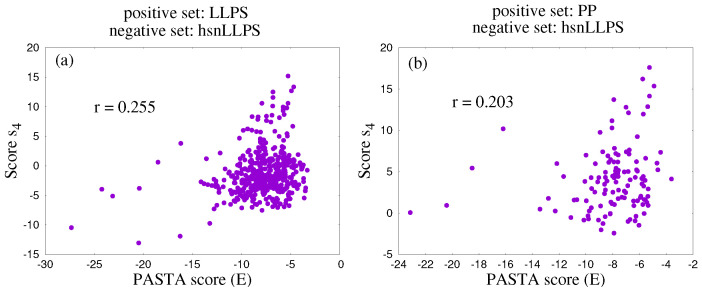
Correlation plot between the PASTA energy *E* and the score $s_{4}$. Each point represents a sequence in the positive set. (**a**) LLPS positive set. (**b**) PP positive set.

**Table 1 biomolecules-12-01771-t001:** Results of one-way ANOVAs to check statistical significance between AUC values on the test set for different groups of scoring functions, with LLPS as positive set and hsnLLPS as negative set.

Scores Tested	*F*	*p*	$\eta^{2}$
$s_{0}$, $s_{00}$	F(1,242)=90.86	<10${}^{-6}$	0.273
Pscore, $s_{1}$, $s_{2}$	F(2,372)=55.46	<10${}^{-6}$	0.23
Pscore, $s_{2}$	F(1,248)=0.018	0.893	∼10${}^{-5}$
$s_{1}$, $s_{3}$	F(1,246)=0.855	0.356	0.003
$s_{1}$, $s_{3}$, $s_{4}$	F(2,366)=15.13	<10${}^{-6}$	0.076
$s_{2}$, $s_{3}$, $s_{4}$	F(2,366)=117.6	<10${}^{-6}$	0.391

**Table 2 biomolecules-12-01771-t002:** Typical values of AUC and MCC on the test set using LLPS as the positive set and hsnLLPS as the negative set. The optimised parameters α, β and γ are also summarised here ($\tilde{\beta }$ and $\tilde{\gamma }$ are actually reported for scores $s_{0}$ and $s_{00}$). The numerical entries are the mean or the median (depending on whether the quantity follows a normal distribution or not, respectively, see Appendix A) ± the standard deviation of the corresponding quantity.

Score	AUC	MCC	α	β	γ
PScore	0.75±0.02	0.26±0.05			
$s_{0}$	0.69±0.04	0.107±0.015		−0.002±1.1	
$s_{00}$	0.71±0.02	0.120±0.014		−0.0030±0.0004	−0.0070±0.0008
$s_{1}$	0.78±0.02	0.26±0.05	120±15		
$s_{2}$	0.75±0.02	0.26±0.05		−0.14±0.05	
$s_{3}$	0.78±0.02	0.26±0.05	120±13	−0.40±0.08	
$s_{4}$	0.79±0.02	0.27±0.04	86±11	−0.56±0.08	−1.40±0.12

**Table 3 biomolecules-12-01771-t003:** Results of one-way ANOVAs to check statistical significance between AUC values on the test set for different groups of scoring functions, with PP as positive set and hsnLLPS as negative set.

Scores Tested	*F*	*p*	$\eta^{2}$
$s_{0}$, $s_{00}$	F(1,246)=0.001	0.975	0
Pscore, $s_{1}$, $s_{2}$	F(2,370)=39.16	<10${}^{-6}$	0.175
Pscore, $s_{1}$	F(1,248)=12.08	6 × 10${}^{-4}$	0.046
$s_{1}$, $s_{3}$, $s_{4}$	F(2,370)=28.04	<10${}^{-6}$	0.132
$s_{2}$, $s_{3}$, $s_{4}$	F(2,368)=0.447	0.64	0.002

**Table 4 biomolecules-12-01771-t004:** Typical values of AUC and MCC on the test set using PP as the positive set and hsnLLPS as the negative set. The optimised parameters α, β and γ are also summarised here ($\tilde{\beta }$ and $\tilde{\gamma }$ are actually reported for scores $s_{0}$ and $s_{00}$). The numerical entries are the mean or the median (depending on whether the quantity follows a normal distribution or not, respectively, see Appendix A) ± the standard deviation of the corresponding quantity.

Score	AUC	MCC	α	β	γ
PScore	0.78±0.05	0.25±0.07			
$s_{0}$	0.75±0.04	0.10±0.03		0.3±1.1	
$s_{00}$	0.75±0.04	0.10±0.03		0.7±0.8	−0.001±0.003
$s_{1}$	0.80±0.04	0.25±0.07	110±18		
$s_{2}$	0.83±0.02	0.25±0.07		1.0±0.1	
$s_{3}$	0.83±0.04	0.25±0.07	40±8	1.0±0.1	
$s_{4}$	0.83±0.04	0.26±0.07	32±11	1.0±0.1	−0.33±0.16

**Table 5 biomolecules-12-01771-t005:** Intrinsic disorder prediction for different sequence sets used in this work. The 3.10.0 release of the MobiDB-lite consensus predictor was used [38]. The sets ${\tilde{S}}_{4}$, ${\tilde{S}}_{P}$, $S_{4}$, $S_{P}$ were defined in Section 3.5, such that ${\tilde{S}}_{4}\subset S_{4}$, ${\tilde{S}}_{4}\cap S_{P}=\emptyset$, ${\tilde{S}}_{P}\subset S_{P}$, ${\tilde{S}}_{P}\cap S_{4}=\emptyset$. hp=humanproteome; mp=mixedproteomes; TP=truepositives.

Sequence Set	Number of Sequences	Total Disorder Fraction	Disordered Segment Mean Length
nsLLPS (mp neg. set)	3503	0.03	37
hsnLLPS (hp neg. set)	16360	0.14	61
LLPS (positive set)	442	0.27	78
$S_{4}$ ($s_{4}$ True Positives)	319	0.30	80
${\tilde{S}}_{4}$ ($s_{4}\;\mathrm{TP}\setminus \mathrm{PScore}\;\mathrm{TP}$)	44	0.14	52
$S_{P}$ (PScore True Pos.)	296	0.32	82
${\tilde{S}}_{P}$ ($\mathrm{PScore}\;\mathrm{TP}\setminus s_{4}\;\mathrm{TP}$)	21	0.22	63

## Data Availability

The data presented in this study are openly available in Zenodo at http://10.5281/zenodo.5778298, reference number 5778299.

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
