# Peer review of "Sequence-Based Prediction of Protein Phase Separation: The Role of Beta-Pairing Propensity"

_biomolecules, 2022, doi:10.3390/biom12121771_

Round 1
Reviewer 2 Report
Please see uploaded 2-page referee report.

Round 2
Reviewer 1 Report
Second review report – Sequence-Based prediction of protein phase separation: the role of beta-pairing propensity.
The paper has improved in providing with biophysical insights about the sets and the relevance of the new score compared to the PScore. I really appreciate this. In several cases, I had to re-read the paragraphs multiple times, especially in those sections that deal with the biophysical interpreation of the results (examples below). I suggest that the authors make an effort to provide clear statements and references and avoid over-interpretation of their model, which is based on sequence information. The discussion section was already long in the first version of the manuscript. It is now even longer. Please consider summarizing the discussion some, which can be achieved by avoiding the many repetitions and over-interpreting the results.
1. I think figure S2 is mis-referenced in the MS. Panel a does not report the distribution of the beta parameter as stated.
2. Paragraph starting at 451: I do not understand why the authors mention that the distributions of the beta parameters are similar. It’s visually clear that the distributions are not similar to each other.
3. Figure S2: since the authors discuss this figure quite extensively, I recommend to draw the distribution of any given parameter, across the different sets, with the same x-axis. For example, panels a, d, g, j should have the x axis from 0 to 140. Likewise for all other columns. If I compare the red distribution (s4 function) of the alpha parameter across panels a, d, g and j, they are quite different. Why? I am afraid that this is not explained in the text or I have missed it. Are these differences reflecting different features of the positive/negative set combinations? Do the authors expect the distribution of s4 across a, d, g, j panels to be the same?
4. Lines 494 to 504: this sounds quite a long explanation of a feature (the position f of a hydrophobic stretch in a sequence) that I do not understand the relevance of. Probably f is not the only quantity that show differences in different sets. Why did the authors pick this quantity? How is this relevant to the paper, whether in terms of technical results and/or biophysical interpretation?
5. Lines 511 to 515: A citation supporting the statement about the different content of disordered protein regions in different proteomes is needed.
6. Why did the authors choose Espritz? It is known that disorder predictors do not always agree. Did the authors check more than one predictor?
7. Statement at line 564: “How protein condensates can be protected…”. What should they protected from?
8. The discussion section is overly long, full of details that are not directly connected with the results and contains repetitions, such as for example lines 603-613, 696-704 and the discussion about the parameters and their distributions, which was already discussed in detail in the Results. Please consider summarizing the discussion section.
9. There is an unreasonably large amount of word in “ ” from line 677 till the end. I admit I was not able to follow the end of the discussion section. Several statements seem speculative and I would avoid over-interpreting the results, since in the end, the model presented by the authors is based exclusively on sequences and scoring functions.
Reviewer 2 Report
Authors addressed my previous concerns adequately. Publication of the revised manuscript recommended.
Round 3
Reviewer 1 Report
I recommend publication of the manuscript, as it has gained in clarity and scope.